# Beyond Pass@$k$: Breadth-Depth Metrics for Reasoning Boundaries

## Abstract

Reinforcement Learning with Verifiable Rewards (RLVR) has emerged as a powerful paradigm to improve Large Language Models on reasoning tasks such as coding, math or logic. To assess the reasoning boundary (the fraction of problems a model can solve) researchers often report Pass@$k$ at large sampling budgets. Recent results reveal a crossover phenomenon: while RLVR models outperform the base model at small k values, the base model usually outperforms them when sampling a very large number of completions. This has been interpreted as evidence that base models have a larger reasoning boundary. We argue that on tasks with discrete answer spaces, such as math with numeric outputs, Pass@$k$ at large k reflects the increasingly higher chance of success in the limit of the number of trials rather than genuine reasoning, and can therefore be misleading. We propose Cover@$\tau$, which measures the fraction of problems that a model can solve for which at least a $\tau$ proportion of completions are correct. Unlike Pass@$k$, Cover@$\tau$ captures reasoning under an explicit reliability threshold: models that rely on random guessing degrade rapidly as $\tau$ increases. We evaluate several RLVR models using Cover@$\tau$-based metrics and illustrate how the relative rankings of popular algorithms change compared to Pass@1, offering a different perspective on reasoning boundaries.

## 1 Introduction

Reinforcement Learning with Verifiable Rewards (Guo et al., 2025) has become an essential post-training approach for improving the capability of LLMs on math, code and logical reasoning. Recent research (Wang et al., 2025b; Wu et al., 2025a) has called into question the extent to which the RLVR models truly expand the reasoning boundary (i.e., extending the scope of solvable tasks). These works report Pass@$k$ for increasing values of k and the reasoning boundary is defined as Pass@$k$ at large k. While the RLVR model has higher Pass@1, at large k, the base model eventually outperforms it, thus the reasoning boundary of the base model shrinks as a result of applying RLVR. Follow-up works use this crossover plot to illustrate whether RLVR expands the reasoning boundary, showing that it can expand or shrink depending on the domain (Liu et al., 2025a; Cheng et al., 2025).

For tasks with numerical answers, such as math, Pass@$k$ at large k may eventually produce correct answers for all problems, due to random guessing rather than reasoning ability. We thus argue that Pass@$k$ can be problematic as a measure for the reasoning boundary, because it doesn't account for any level of reliability, for any given problem. We thus complement it with a *reliability-controlled reasoning boundary*, that exposes the reliability level explicitly. Concretely, we define Cover@$\tau$, closely related to G-Pass@K (Liu et al., 2024), which measures the fraction of problems solved by a model with success rate at least $\tau$ without relying on setting a K parameter. For very small $\tau$, Cover@$\tau$ behaves similarly to Pass@$k$, however, increasing $\tau$ tightens the criterion to emphasize consistency over chance. Figure 1 highlights the behavior of Pass@$k$ and Cover@$\tau$ for a math dataset from OMEGA (Sun et al., 2025) with numerical answers and small set support.

Our contributions are as follows:

- We adapt the idea of reliability-thresholded evaluation to analyze reasoning performance at varying reliability levels $\tau$. Specifically, we define Cover@$\tau$, which, in contrast to Pass@$k$, highlights an explicit breadth-depth trade-off: low $\tau$ captures breadth of problem solving (even if unreliable), while larger $\tau$ captures depth (problem solving with high reliability). Measuring Cover@$\tau$ reveals a different ranking of popular RLVR algorithms

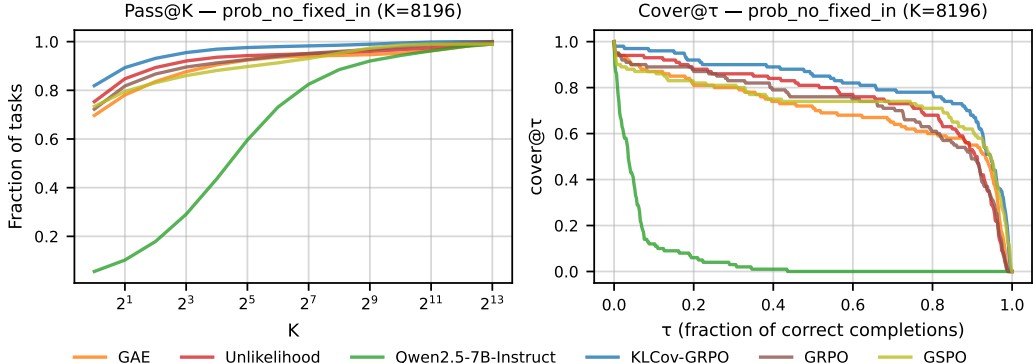

Figure 1: Pass@$k$ and Cover@$\tau$ curves for Qwen2.5-7B-Instruct and several RLVR models on the Probability set of OMEGA. **Left:** Pass@$k$ quickly saturates for larger $k$ due to small test support. **Right:** Cover@$\tau$ illustrates a more gradual assessment of the models' capabilities, ranging from maximum performance (at low $\tau$ values) to very limited capabilities (when requiring models have almost perfect reliability at high $\tau$ values).

when compared to Pass@1 or Pass@$k$ at large k; this shows a complementary perspective of the model capabilities

- we demonstrate that Pass@$k$ is a weighted average of Cover@$\tau$, with weights from a Beta(1,k) distribution; this reveals that Pass@$k$ is biased towards low-$\tau$ regions of Cover@$\tau$, emphasizing lucky hits rather than reliability

- we illustrate the usefulness of Cover@$\tau$ by characterizing the performance of different RLVR methods trained on math datasets from OMEGA and Reasoning Gym.

## 2 RELATED WORK

DeepSeekR1 (Guo et al., 2025) has paved the way for reasoning-focused LLMs, by finetuning LLMs with reinforcement learning from verifiable rewards. The model employs the Group Relative Policy Optimization (GRPO) algorithm (Shao et al., 2024), which is a critic-free variant of PPO (Schulman et al., 2017), that estimates the baseline from the group average rewards.

Our work is motivated by Yue et al. (2025), which has sparked significant discussion about whether RL-based fine-tuning truly enhances the underlying reasoning abilities of large language models rather than simply increasing the sampling efficiency of existing reasoning paths.

### 2.1 EVALUATION METRICS

Models are usually evaluated via Pass@$k$ (Chen et al., 2021), using $k = 1$ as the most common value. Higher values of $k$ (e.g. 16 or 64) are also used to test the upper bound capabilities of the models. Recent work argues that the relative decrease in reasoning capacity of RLVR models compared to the base model is an artefact of Pass@$k$ and proposes CoT-Pass@k (Wen et al., 2025) to account for the correctness of both the thinking tokens and the final answer. Finally, some papers report maj@k or cons@k (Guo et al., 2025; Shao et al., 2024), which counts a problem as solved based on aggregation over k samples, either majority (>50% of completions correct) or mode (most frequent). Both metrics target whether a model is consistent and are related to our proposal. Maj@k is equivalent to Cover@$\tau$ with $\tau = 0.5$. Cons@k has no fixed reliability threshold: the effective $\tau$ varies per problem with the mode's frequency, so it does not correspond to a single Cover@$\tau$. Our Cover@$\tau$ is structurally similar to the performance profiles advocated by Agarwal et al. (2021), where the fraction of solved games is reported relative to varying human-normalized score levels. The key distinction is we evaluate coverage under varying reliability levels using a fixed (and sufficiently large) sample budget of $k$ completions, without normalizing to human performance.

Liu et al. (2024) et al. introduce G-Pass@$k_\tau$, a generalization of Pass@$k$ that evaluates the capability of the model under different consistency levels $\tau$. Our metric Cover@$\tau$ was developed independently and, while conceptually similar, differs in formulation and intended use. First, our metric assumes that tasks have different per-trial success probabilities $p_i$, estimated directly from the n completions of each task, eliminating the need to set a separate k parameter. Second, we introduce our metric in the context of the reasoning boundary debate, whereas Liu et al. (2024) use G-Pass@$k_\tau$ primarily to assess model-level stability under repeated sampling. To this end, G-Pass@$k_\tau$ focuses on higher reliability levels ($\tau \in [0.5, 1.0]$), while we use Cover@$\tau$ to analyze the breadth-depth trade-off, by evaluating performance at both low ($\tau = 0.2$) and high ($\tau = 0.8$) reliability thresholds.

## 2.2 EXPLORATION-PROMOTING METHODS

Several shortcomings in GRPO have been reported, notably optimization biases (Liu et al., 2025b; Zheng et al., 2025) and entropy collapse (Yu et al., 2025). The latter is particularly problematic, because it reduces exploration and quickly saturates performance (Cui et al., 2025). Follow-up work addresses the entropy collapse and encourages exploration. Simpler approaches add an explicit entropy loss term (Wang et al., 2025b) or increase the upper clipping threshold to favor low-probability exploration tokens (Yu et al., 2025). More fine-grained techniques such as KL-cov (Cui et al., 2025) suppress the high-covariance tokens that correlate with large decreases in entropy. Other works identify forking tokens (Wang et al., 2025a), which are high entropy tokens serving as logical connectors. Restricting policy gradient updates to these high-entropy tokens maintains entropy and enhances exploration. Finally, GRPO-Unlikeliness (He et al., 2025) promotes low-probability solutions by penalizing the reward of high-probability ones.

## 2.3 PROCEDURALLY GENERATED DATASETS

Progress of RLVR methods on math reasoning has been questioned, due to potential contamination in popular static benchmarks. Portion of these datasets may appear in the pretraining data of popular LLMs (Wu et al., 2025b), which can conflate reasoning abilities with memorization. To limit contamination concerns, recent math and logic datasets are procedurally generated and designed with increasing difficulty and structural variation (Sun et al., 2025; Stojanovski et al., 2025).

## 3 PASS@$k$ CAN BE MISLEADING

We consider $T$ tasks, where task $i$ has per-trial success probability $p_i \in [0, 1]$ under i.i.d. trials. Pass@$k$ is defined as the average probability that a task is solved within k independent attempts.

**Definition 1** (Pass@$k$)**.**

$$Pass@\mathrm{k} \;=\; \frac{1}{T} \sum_{i=1}^{T} \left( 1 - (1 - p_i)^k \right).$$

Pass@$k$ has been used to assess the reasoning boundaries of RLVR models by comparing the performance of a finetuned model against the base model at varying values of $k$. At large $k$ values, the performance of the base model typically approaches or even surpasses that of the finetuned model, seemingly closing any gap that may exist at small $k$ values. This pattern, however, does not necessarily reflect genuine reasoning ability. Instead, it primarily highlights the diversity of output trajectories in the base model. In the mathematical reasoning setting, where tasks often involve numerical answers with very limited support, this effect can create a misleading impression about model reasoning.

Figure 1 illustrates this point by plotting Pass@$k$ curves for the base model and several RLVR variants on the Probability (No Fixed) task from the OMEGA dataset. Given enough trials ($2^{13}$), the base model achieves Pass@$k = 1$, due to the very limited size of the output space (only 30 possible values in the test set). More generally, this phenomenon is inevitable: provided there is nonzero probability of producing the correct answer, Pass@$k$ will always converge to 1 in the limit of infinite trials.

**Remark 1** (Degeneracy of Pass@$k$ at Large $k$)**.** *For any success probability* $0 < p \leq 1$,

$$\lim_{k \to \infty} \left( 1 - (1 - p)^k \right) = 1.$$

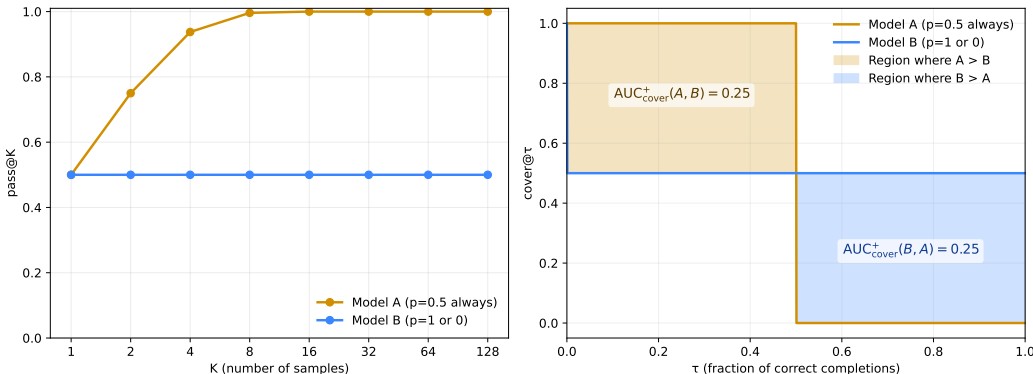

Figure 2: **Left:** pass@K for two models A and B. Both have the same pass@1=0.5, but model A's reasoning boundary increases with more tries, while model B stay flat. **Right:** Cover@$\tau$ curves for the same models A and B. Model A solves more problems overall, while Model B solves fewer problems but with higher consistency. When comparing their excess AUC (areas where each curve dominates), their overall advantages balance out.

We thus argue that using Pass@$k$ as a proxy for reasoning boundaries can be misleading, because it confounds true capability with random chance.

## 4    COVER@$\tau$

Whereas Pass@$k$ captures the binary likelihood of success within $k$ attempts, we propose examining how dataset coverage changes when accounting for the consistency of predictions. We define Cover@$\tau$ as fraction of problems for which at least a proportion $\tau$ of the generated completions are correct. Formally:

**Definition 2** (Cover@$\tau$). *For any threshold $\tau \in [0, 1]$, define*

$$G(\tau) \; = \; \frac{1}{T} \sum_{i=1}^{T} \mathbf{1}\{p_i \geq \tau\},$$

*i.e. the fraction of tasks that can be solved with per-trial success probability at least $\tau$.*

Consider two LLMs, tested on the same set of problems: A has probability of success of 0.5 on all problems, while B has probability of success 0 on half the problems and 1 on the other half. Both models have $Pass@1 = 0.5$, but the **Cover@$\tau$** plot in the right side of Figure 2 also shows model performances at explicit reliability levels: model A solves more problems, while model B solves fewer problems, but more consistently. Thus, Cover@$\tau$ captures a fine-grained view of the performance regime of the models:

- low $\tau$ values: Cover@$\tau$ highlights the **breadth** of the capabilities (how many problems are at least sometimes solvable)
- higher $\tau$ values: Cover@$\tau$ indicates **depth** (how reliably a problem is solved)

Two models A and B can cross over in their Cover@$\tau$ curves: model A performs better at lower reliability thresholds, but B dominates at higher thresholds. To capture such cases where models trade dominance at different $\tau$ levels, we define $\text{AUC}_{cover}^{+}(A, B)$. This is the excess AUC between model's A coverage over model's B coverage across reliability thresholds:

$$\text{AUC}_{cover}^{+}(A, B) = \int_{0}^{1} \max\big(G_A(\tau) - G_B(\tau), \, 0\big) \, d\tau,$$

where $G_M(\tau)$ denotes the Cover@$\tau$ curve of model $M$. This pairwise metric captures the total coverage advantage of model A relative to model B, ignoring regions where coverage of A is worse. In the right side of Figure 2, both models have equal $\text{AUC}_{cover}^{+}$, indicating a similar level of performance.

## 5  RELATING PASS@$k$ AND COVER@$\tau$

We now demonstrate the connection between Pass@k and Cover@$\tau$. Specifically, Pass@$k$ is a Beta-weighted average of the Cover@$\tau$ metric, and thus represents only one particular projection of the richer information contained in Cover@$\tau$.

**Proposition 1.** *Pass@$k$ as Weighted Average of Cover@$\tau$.*

*For any $k \geq 1$,*

$$Pass@\mathrm{k} = \int_0^1 k(1-\tau)^{k-1}\, G(\tau)\, d\tau,$$

*where $G(\tau)$ denotes the Cover@$\tau$ curve.*

*Proof.* For a single task with success probability $p$, define $f_k(p) = 1 - (1-p)^k$. Since $f_k'(p) = k(1-p)^{k-1}$ and $f_k(0) = 0$, the fundamental theorem of calculus gives

$$f_k(p) = \int_0^p f_k'(\tau)\, d\tau = \int_0^1 k(1-\tau)^{k-1}\, \mathbf{1}\{p \geq \tau\}\, d\tau.$$

Averaging over tasks,

$$\mathrm{Pass@}k = \frac{1}{T}\sum_{i=1}^T f_k(p_i) = \int_0^1 k(1-\tau)^{k-1}\left(\tfrac{1}{T}\sum_{i=1}^T \mathbf{1}\{p_i \geq \tau\}\right) d\tau.$$

The term in parentheses is exactly $G(\tau)$, which yields the claim. □

**Corollary 1.** *Pass@$k$ as Expectation under Beta Weights.*

*Let $\tau \sim \mathrm{Beta}(1, k)$ with density $p_k(\tau) = k(1-\tau)^{k-1}$ on $[0, 1]$. Then*

$$Pass@k = \int_0^1 G(\tau)\, p_k(\tau)\, d\tau = \mathbb{E}_{\tau \sim \mathrm{Beta}(1,k)}[\, G(\tau)\,].$$

**Corollary 2.** *Uniform AUC Equals Pass@1.*

*The unweighted area under the Cover@$\tau$ curve satisfies*

$$\int_0^1 G(\tau)\, d\tau = \frac{1}{T}\sum_{i=1}^T p_i = \mathrm{Pass@1}.$$

**Remark 2.** *As $k \to \infty$, the weighting distribution $\mathrm{Beta}(1, k)$ concentrates at $\tau = 0$. Therefore*

$$\lim_{k \to \infty} Pass@k = G(0^+),$$

*the fraction of tasks with nonzero success probability. If every $p_i > 0$, then Pass@$k \to 1$.*

**Proposition 2.** *Cover@$\tau$ dominance implies Pass@$k$ dominance.*

*Given two models A and B, if Cover@$\tau(A) \geq$ Cover@$\tau(B)$ for all $\tau \in [0, 1]$, then Pass@$k(A) \geq$ Pass@$k(B)$ for every $K \geq 1$.*

*Proof.* Using proposition 1,

$$\mathrm{Pass@k}(A) - \mathrm{Pass@k}(B) = \int_0^1 k(1-\tau)^{k-1}\,(G_A(\tau) - G_B(\tau))\, d\tau,$$

Since $k(1-\tau)^{k-1} \geq 0$ and $G_A(\tau) - G_B(\tau) \geq 0$, the integral is positive. Therefore Pass@k$(A) \geq$ Pass@k$(B)$. □

### OBSERVATIONS

**Pass@$k$ as a weighted summary of Cover@$\tau$.**  Proposition 1 shows that Pass@$k$ is a weighted average of the Cover@$\tau$ curve, where the weights are given by a $\mathrm{Beta}(1, k)$ distribution. In other words, Pass@$k$ summarizes only part of the information already contained in Cover@$\tau$.

**Bias toward low thresholds.** The $\text{Beta}(1, k)$ weighting heavily emphasizes small values of $\tau$ as $k$ grows. Thus Pass@$k$ primarily captures whether tasks have *any* nonzero success probability, rather than how reliably they can be solved. In the limit $k \to \infty$, Pass@$k$ collapses to the trivial statistic "fraction of tasks with $p_i > 0$."

**Uniform weighting recovers Pass@1.** When $G(\tau)$ is weighted uniformly, the resulting area under the Cover@$\tau$ curve equals Pass@1, i.e. the average per-trial success probability. This highlights that Cover@$\tau$ naturally generalizes both Pass@1 and Pass@$k$.

**Cover@$\tau$ ranking is more informative than Pass@k** Proposition 2 shows that rankings based on Cover@$\tau$ imply the corresponding rankings based on Pass@k. However, the reciprocal is not true, as evidenced by Figure 2. Model A outperforms model B in terms of Pass@k for all K, but the Cover@$\tau$ curve reveals ordering differences across reliability levels that the Pass@k curve can hide.

**Summary.** Cover@$\tau$ makes the coverage-reliability trade-offs explicit, avoids the degeneracy of large-$k$ behavior, and reveals finer-grained rankings that Pass@k can obscure.

# 6 RE-ASSESING GENERALIZATION PERFORMANCE ON MATH USING COVER@TAU

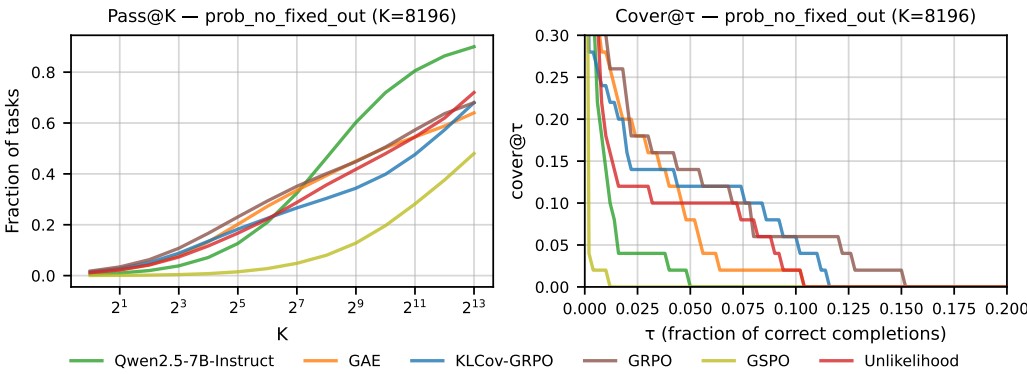

Figure 3: Pass@k and Cover@$\tau$ curves for Qwen2.5-7B-Instruct and RLVR models on the Probability (No Fixed) subset of OMEGA, for the OOD test split. **Left:** All models have poor accuracy, and increasing the sampling budget leads to higher Pass@$k$, especially on the base model. **Right:** GRPO and KL-Cov generalize the best; the base model quickly drops in performance even at low reliability thresholds, suggesting a far more limited reasoning boundary than the $Pass@k$ plot implies.

We evaluate RLVR methods using Cover@$\tau$-based metrics, focusing on: (i) highlighting the trade-offs exposed by different $\tau$ values (ii) examining how models perform in mathematical reasoning under OOD settings.

## 6.1 EXPERIMENTAL SETUP

**Datasets** Popular datasets for evaluating mathematical reasoning may suffer from data leakage (Wu et al., 2025b). As a result, training on these datasets may not accurately reflect improvements relative to the base model. We instead focus on two particular datasets: OMEGA (Sun et al., 2025) and Reasoning Gym (Stojanovski et al., 2025). OMEGA provides both IID and OOD test splits across various tasks, from which we select: GCD, Function Intersection, Probability (No Fixed), Digit Sum, and Circle. For all tasks, we train on IID data and test on the OOD splits. For Reasoning Gym, we follow the intra-domain transfer setup: we train on two tasks from the algebra domain (simple equations and polynomial multiplication) and test on a third (intermediate integration).

**Methods** We evaluate two commonly used RLVR algorithms: GRPO (Shao et al., 2024) and PPO (Schulman et al., 2017). In addition, we test GSPO (Zheng et al., 2025), which improves upon

GRPO by defining importance ratios at the sequence level rather than the token level. Given our out-of-distribution setup, we also consider methods specifically designed to enhance exploration. KL-Cov (Cui et al., 2025) applies a KL penalty to tokens to tokens with high covariance between their log probabilities and advantages, helping to prevent entropy collapse, while GRPO-Unlikeliness (He et al., 2025) introduces an unlikeliness reward, which incentivizes correct but less probable solutions.

**Training details**   We train all models using the VERL framework (Sheng et al., 2024), starting from the Qwen-2.5-7B-Instruct model (Yang et al., 2025). For the Omega tasks, we run training for 30 epochs with a batch size of 500 and a mini-batch size of 96. For Reasoning Gym, we train for 5 epochs, with a batch size of 64 and a mini-batch size of 32. We set PPO epochs to 1 for all experiments. The actor and critic use learning rates of $1e-6$ and $1e-5$, respectively. For the KL loss between the policy and base model, we use the following: 0.1 on the Probability (No Fixed) task, as well as on Reasoning Gym for the GRPO-Unlikeliness method, 0.05 for the other Omega tasks, and 0.01 for all other Reasoning Gym experiments. Across all methods, we use 32 (OMEGA) and 8 (Reasoning Gym) rollouts per prompt which, for GRPO-style methods, corresponds to the group size. We select the best performing checkpoint on the iid validation sets for evaluation. All experiments were performed on a single node with 8 NVIDIA H200 GPUs.

## 6.2   MODEL ANALYSIS USING THE COVER@$\tau$ CURVE

The right side of Figure 1 illustrates the Cover@$\tau$ on the IID split of the Probability (no fixed) subset of OMEGA. Our metric highlights the brittleness of the base model: while it solves problems at very small thresholds, its coverage quickly decreases even at modest thresholds, such as $\tau = 0.2$. KL-Cov consistently solves the most problems at almost all the reliability thresholds.

In Figure 3 we plot both the Pass@k and Cover@$\tau$ curves for the OOD split of the same dataset. All models have poor Pass@1 performance, but, for large sampling budgets, the base model significantly outperforms the RLVR models. For very small values of $\tau < 0.01$, Cover@$\tau$ highlights a similarly strong performance of the base model. This correlates with the theoretical insights from Section 5, showing that as $\tau$ approaches 0, it becomes similar to Pass@k at large $k$s. However, when marginally increasing the threshold to $\tau = 0.025$, the performance of the base model drops significantly, which showcases a less optimistic view of its reasoning capabilities. Additionally, the Cover@$\tau$ curve reveals different trade-offs between the RLVR models. GAE solves more problems than GRPO-Unlikeliness for $\tau < 0.050$, while GRPO-Unlikeliness is more reliable for larger $\tau$ values.

**Cover@$\tau$ offers finer granularity.**   By inspecting the entire curve, we can distinguish between algorithms that (a) succeed rarely across many tasks, and (b) succeed reliably on fewer tasks. Pass@k targets the former, while being uninformative about the latter. Cover@$\tau$ exposes important trade-offs between *coverage* and *reliability* of exploration.

## 6.3   RESULTS

In Table 1 we evaluate several RLVR algorithms on the Reasoning Gym and OMEGA datasets. We report pass@1, as well as Cover@$\tau$ for $\tau = 0.2$ and $\tau = 0.8$, to assess both low and high reliability performance. While these point metrics provide insight into the performance at explicit reliability

Table 1: Results on the Intermediate Integration task from Reasoning Gym and the OMEGA benchmark (averaged across 5 tasks). Best results per metric are shown in bold, second and third best are highlighted in blue and orange, respectively.

| Method | Reasoning Gym | | | | OMEGA OOD | | | |
|---|---|---|---|---|---|---|---|---|
| | Pass@1 | cov@.2 | cov@.8 | $\text{AvgAUC}_{\text{cov}}^{+}$ | Pass@1 | cov@.2 | cov@.8 | $\text{AvgAUC}_{\text{cov}}^{+}$ |
| base | 49.67 | 55.67 | 39.67 | 0.64 | 8.34 | 14.94 | 1.10 | 0.19 |
| GRPO | **59.66** | **66.00** | 48.67 | 5.52 | 17.86 | 22.66 | 11.98 | 1.61 |
| GSPO | 56.14 | 57.67 | 48.00 | 1.79 | 18.00 | 20.26 | 15.36 | 2.36 |
| PPO (GAE) | 57.94 | 60.00 | 55.67 | 5.13 | 18.38 | 22.66 | 13.68 | 1.85 |
| KL-Cov | 58.55 | 62.00 | **56.33** | **5.70** | **28.34** | **33.58** | **23.34** | **12.78** |
| Unlikeliness | 43.98 | 51.67 | 34.00 | 0.00 | 17.02 | 20.94 | 11.94 | 0.68 |

levels, they do not capture how models perform across the entire range of reliability thresholds. To summarize a model's performance across all $\tau$ regions, we build on the previously defined pairwise measure $\mathrm{AUC}^+_{\mathrm{cover}}(A, B)$. Consider a set of models $\mathcal{M}$, with $|\mathcal{M}| = M$. For a model A, we average its pairwise excess AUC score against all the other models:

$$\mathrm{AvgAUC}^+_{\mathrm{cover}}(A) \;=\; \frac{1}{M-1} \sum_{\substack{B \in \mathcal{M} \\ B \neq A}} \mathrm{AUC}^+_{\mathrm{cover}}(A, B) \tag{1}$$

$\mathrm{AvgAUC}^+_{\mathrm{cover}}(A)$ quantifies how much A tends to dominate other models on the Cover@$\tau$ curve.

In Table 1 we observe that the rankings of the top methods differ between Pass@1, Cover@$\tau$ and $\mathrm{AvgAUC}^+_{\mathrm{cover}}$. While cover@0.2 yields the same rankings as Pass@1, with GRPO as the top method, cover@0.8 ranks KL-cov first, highlighting its strength in preserving depth at higher thresholds. The $\mathrm{AvgAUC}^+_{\mathrm{cover}}$ score (1) averages performance across dominant threshold regions: KL-Cov is ranked first, while GRPO ranks second.

KL-Cov obtains the best results on the OMEGA dataset on all metrics. While GRPO ranks 4th based on Pass@1, it ranks 2nd based on Cover@0.2 (tied with PPO), highlighting better coverage at low reliability. Enforcing higher reliability ($\tau = 0.8$) produces a different ordering compared to $\tau = 0.2$, where breadth is emphasized. GRPO and GRPO-Unlikeliness show good Cover@$\tau$ performance at lower thresholds, but their coverage at $\tau = 0.8$ drops out of the top 3. The $\mathrm{AvgAUC}^+_{\mathrm{cover}}$ score (1) captures this trade-off: GRPO and GRPO-Unlikeliness lag behind GSPO and PPO, which are ranked 2nd and 3rd, respectively. Overall, the strong performance of KL-Cov suggests that RLVR methods that prevent entropy collapse show stronger generalization abilities.

## 7 LIMITATIONS

Our metric still accounts for the accuracy of the final answer, without evaluating the soundness of the reasoning trace, as was done in Cot-Pass@k(Wen et al., 2025). However, the accuracy of the reasoning trajectory can be combined with the Cover@$\tau$ metric.

## 8 CONCLUSIONS

We adapt the idea of reliability-thresholded metrics and define Cover@$\tau$, which emphasizes a *reliability-controlled reasoning boundary*. This fine-grained view of the performance highlights the trade-off between the coverage of solvable problems and the correct answer consistency. We connect Cover@$\tau$ to Pass@k, demonstrating that Pass@$k$ can be expressed as a weighted average of Cover@$\tau$ with the weights given by a $\mathrm{Beta}(1, k)$ distribution. Thus, Pass@k is biased toward low $\tau$ regions of Cover@$\tau$, making it prone to emphasizing lucky hits rather than reliability. We also connect Cover@$\tau$ to Pass@1, which is the AuC of Cover@$\tau$. Through this new lens, we evaluate several RLVR methods to uncover their reasoning abilities under different reliability thresholds. Preliminary results demonstrate the advantage of entropy-based methods (KL-Cov) with respect to their generalization abilities.

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
