# OpenReview forum: "Beyond Pass@k: Breadth-Depth Metrics for Reasoning Boundaries"
_ICLR.cc/2026/Conference — Submitted to ICLR 2026_

### Official Review · Reviewer_uWMw · 2025-10-14

**Soundness:** 3
**Presentation:** 3
**Contribution:** 2
**Rating:** 4
**Confidence:** 4

**Summary:**

This paper introduces Cover@τ, a new evaluation metric intended to complement or replace Pass@k when assessing the reasoning performance of Reinforcement Learning with Verifiable Rewards (RLVR) models. The authors argue that Pass@k can be misleading for tasks with small discrete answer spaces, such as mathematical reasoning, because increasing k artificially raises success rates through random chance rather than genuine reasoning ability. They formally define Cover@τ as the fraction of tasks for which at least a τ fraction of completions are correct, show that Pass@k is a Beta-weighted integral of Cover@τ, and empirically evaluate various RLVR models (GRPO, GSPO, PPO-GAE, KL-Cov, GRPO-Unlikeliness) on OMEGA and Reasoning Gym datasets. The results suggest that Cover@τ provides a more nuanced view of “breadth” versus “depth” of reasoning capabilities, favoring exploration-preserving algorithms such as KL-Cov.

**Strengths:**

###  Clear problem statement and motivation:

The paper articulates a genuine issue with Pass@k saturation for small answer spaces and provides both intuitive and formal justification.

### Elegant theoretical connection:

The derivation showing Pass@k as a weighted average over Cover@τ via a Beta(1, k) distribution is mathematically neat and offers interpretability.

### Visualization clarity:

The Pass@k vs. Cover@τ plots (Figures 1–3) effectively demonstrate the saturation and trade-off phenomena, supporting the paper’s main claims.

**Weaknesses:**

### Limited novelty beyond metric reformulation:

While Cover@τ is conceptually appealing, it is essentially a restatement of existing majority or consistency metrics (maj@k, cons@k) with a continuous threshold. The main novelty lies in the interpretation, not in the metric itself. This weakens the contribution for a top-tier venue.

### Overstated theoretical contribution:

The mathematical results (Proposition 1–2 and corollaries) are straightforward consequences of calculus and probability theory. They provide insight but do not represent new theory on reasoning evaluation or RLVR optimization.

### ncomplete problem diagnosis:
The paper identifies a real issue with Pass@k but the solution doesn't fully address it:

- The problem occurs specifically when answer spaces are small and uniformly distributed. The paper doesn't systematically characterize when Pass@k is problematic vs. when it's fine.

- For many reasoning tasks (especially open-ended or continuous domains), the random guessing problem is minimal, making the motivation less universal than presented.

- No analysis of task properties that determine whether random guessing is a concern.

### Practical utility unclear:

- The paper shows that rankings can differ across τ values, but doesn't establish when practitioners should care about different τ levels.

- No clear recommendation for which τ to use. Should we report a τ profile or a single number? If a single number, how do we choose it?

- The claim that Cover@τ is "more informative" is not well-supported. Information ≠ utility. Practitioners need guidance on decision-making, not just more curves.

**Questions:**

I know my review is kind of critical. But would be pleased to see your response over:

- In practical applications (e.g., code generation, math competition problems), how often is random guessing actually a problem?

- How should practitioners choose τ for their use case?

- Can you provide guidance on when Cover@τ should replace Pass@k vs. complement it?

---

> ### Author Response · Authors · 2025-11-24
>
> Thank you for reading our work and raising many important issues!
>
> **cover@τ as restatement of existing majority or consistency metrics (maj@k, cons@k) with a continuous threshold**: Maj@k can indeed be viewed as a special case of cover@τ with τ=0.5. However, for this threshold, neither good performance at low τ (breadth) nor good performance at high τ (depth) is captured. Cover@τ introduces a continuous reliability axis τ in [0,1], which disentangles breadth from depth and enables systematic analysis of how models trade off between them.
>
> **pass@k behavior with respect to non-numerical answers and other domains**: While we illustrated the "lucky hits" argument for pass@k for test sets with numerical answers and small support (Omega), we also evaluated a domain with a substantially larger symbolic answer space (Reasoning Gym, where answers are integrals expressions). We included an additional plot where pass@K and cover@τ curves are illustrated on the Reasoning Gym test set (link: https://ibb.co/tM3QFQhY). Here, the crossover persists, though it is less extreme: at large K (4096), the base model is almost on par with the PPO model in terms of pass@K and the plot suggests pass@k increasing beyond K=4096 as well. The cover@τ curves illustrate that the base model has less coverage than almost all the other RLVR methods at both low-τ (breadth) and high-τ (depth) regions. This indicates that the "lucky hits at large k" behavior is not limited to numeric answers with small support, it also appears with moderate sampling budgets in symbolic answer spaces, where the support is larger but still structured.
>
> We agree that experiments for code generation would strengthen the conclusions. Notably, the crossover phenomenon between the base and RLVR models has also been reported in code (Figure 4 in Yue et al. https://arxiv.org/pdf/2504.13837), showing that pass@k inflation with K happens even at K=128, despite the answer space being large.
>
> **How should practitioners choose τ for their use case?**
> We agree that actionable recommendations are important for practitioners.
> We view the full cover@τ curve as an exploration tool: it provides a complete reliability profile and reveals exactly where curves cross between two models.
> Point evaluations provide comparisons at specific reliability levels. We provided τ = 0.2 and τ = 0.8 to assess breadth-depth as they illustrate two ends of the reliability spectrum, but these values are illustrative rather than prescriptive, and practitioners may choose others depending on application demands.
>
> Choosing τ values based on reliability:
> * low τ values (e.g. 0.1-0.3) reflects breadth capabilities ("how many tasks does the model solve at least sometimes?"); these values are also useful for picking models where creativity or partial success is valuable (such as, searching for research ideas)
> * high τ values (e.g. 0.7-0.9) reflects depth ("how many tasks does the model consistently solve?"); these values are also useful for selecting models where there is less room for errors (such as, code generation)
>
> We intentionally refrained from prescribing specific values, as reliability requirements vary across applications. If a single value is required, it should reflect the risk tolerance of the task (lower τ values for problems where errors are tolerated and higher τ values for problems where consistency is key).
>
> **Can you provide guidance on when Cover@τ should replace Pass@k vs. complement it?** We believe cover@tau should complement pass@k, not replace it.
>
> We provide the following recommendations:
> * Applications/practitioners: evaluate cover@τ for low τ values (e.g. 0.1-0.3) or high values (e.g. 0.7-0.9) depending on reliability needs
> * Research on reasoning boundaries: use full curves for exploration and a small number of τ values at both ends (e.g. 0.2 and 0.8) for breadth-depth comparisons
> * Model comparison (both practitioners and researchers):
>   * use pass@1 as a global summary metric
>   * report cover@τ at values relevant for the task's reliability regime
>   * report pairwise metrics such as AvgAuc+ for finer grained comparisons

---

> > ### Comment · Reviewer_uWMw · 2025-11-24
> >
> > Thanks for your quick response. However, I would like to see more empirical results instead of pure text explanations. For instance, for my first question, it is preferred to run some random guess experiments to demonstrate that it is actually a problem, right?

---

> > > ### Author Response · Authors · 2025-11-26
> > >
> > > We ran a random guess baseline and measured its performance on 5 OOD test sets of the Omega dataset: arithmetic_gcd_out, digit_sum_out, circle_out, func_area_out and prob_no_fixed_out.
> > > The random guess predictor samples values uniformly from [min_train_value, max_train_value]. We plot the pass@k and cover@tau for the random guess baseline alongside the base model and RLVR models.
> > >
> > > The plots are listed here:  [prob_no_fixed_out](https://files.catbox.moe/yj1rm6.png), [circle_out](https://files.catbox.moe/fy3yug.png), [func_area_out](https://files.catbox.moe/z2036w.png), [arithmetic_gcd_out](https://files.catbox.moe/rl8itn.png), [digit_sum_out](https://files.catbox.moe/44zfrq.png). We also show a zoomed in plot of cover@tau for [prob_no_fixed](https://files.catbox.moe/hsa16n.png), where we can see that cover@tau quickly decreases to 0 for very small values of tau.
> > >
> > > For the random guess predictor, in 3/5 datasets, pass@K inflates at large K values and cover@tau drops sharply. On the other 2 datasets, pass@k performance shows no improvements due to the OOD test set range being outside the train set range.
> > > These results confirm that using pass@K at large K for assessing reasoning boundaries on math problems with numeric answers is problematic.
> > >
> > > As for tasks with symbolic answers spaces (such as math proofs or code), a random guess predictor will not inflate pass@K. However, pass@K still shows hints of problematic behaviour at large K. Specifically, for math, we evaluated the base and RLVR models on a Reasoning Gym set, where answers are integral expressions ([link to plot](https://ibb.co/tM3QFQhY)). Here, the crossover persists, though it is less extreme: the base model is almost on par with the PPO model in terms of pass@K and the plot suggests pass@k increasing beyond K=4096 as well. This indicates that the "lucky hits at large k" behavior is not limited to numeric answers with small support, it also appears with moderate sampling budgets in symbolic answer spaces, where the base model reaches the correct solution with low probability.

---

> > > > ### Comment · Reviewer_uWMw · 2025-11-26
> > > >
> > > > Thanks for your quick response. Due to the great efforts of the author in addressing my concerns. I would like to increase my score. But I have no opinion on whether the 0-score reviewer is correct or not. I would recommend the author explain more clearly to his/her critics.

---

> > > > > ### Author Response · Authors · 2025-11-27
> > > > >
> > > > > Thank you for your follow-up, recommendations and for reconsidering your score. We appreciate the time you took to read our rebuttal and engage constructively with our work.

---

### Official Review · Reviewer_Nwpw · 2025-10-31

**Soundness:** 3
**Presentation:** 2
**Contribution:** 1
**Rating:** 0
**Confidence:** 5

**Summary:**

This paper introduces Cover@τ, a reliability-thresholded extension of Pass@k that measures the fraction of problems solved with at least a τ proportion of correct completions. The metric is used to analyze reasoning boundaries of RLVR models and slove limitations of Pass@k at large sampling numbers.

**Strengths:**

1. The paper is clearly written, with well-organized sections and consistent notation.
2. The topic of evaluation metrics for reasoning stability is important for RLVR research community.

**Weaknesses:**

The core idea of thresholded generalization of Pass@k is not novel. The recently published ACL 2025 paper “Are Your LLMs Capable of Stable Reasoning?” introduced an almost identical formulation, G-Pass@k and mG-Pass@k, with the same motivation and nearly equivalent equations. That paper also showed that Pass@k is a special case as τ→0 and provided extensive stability analyses.

The current submission does not cite or discuss that prior work, which gives a misleading impression of originality. Mathematically, Cover@τ and G-Pass@kτ differ only in notation. Hence, the contribution appears incremental without proper acknowledgment or comparison.

**Questions:**

As mentioned in weaknesses.

**Details Of Ethics Concerns:**

The submission introduces a metric (Cover@τ) that is conceptually and mathematically equivalent to the G-Pass@k metric proposed in the ACL 2025 paper “Are Your LLMs Capable of Stable Reasoning”, without any citation or acknowledgment. This raises potential research integrity concerns regarding proper attribution of prior work.

---

> ### Author Response · Authors · 2025-11-13
> **Response**
>
> **Clarification on the research integrity issue:**
>
> Thank you for bringing up this issue. After submitting our work to ICLR 2026, we became aware of the work of Liu et al. 2024 (Are your LLMs capable of stable reasoning?), whose metric G-Pass@k is conceptually similar to ours. Our work was developed independently of Liu et al. and we want to clarify how this overlap arose.
>
>
> Our ICLR submission was also uploaded as a preprint manuscript. Once we discovered the prior work, we uploaded an updated preprint version, in order to i) properly cite Liu et al. ii) clarify the overlap but also the difference in formulation and intended use. We started working on this paper in early 2025, where the Cover@τ metric arose in the context of the sharpening vs. discovery debate on RLVR’s role. In contrast, G-Pass@k was introduced earlier and focuses more on model stability. We were thus motivated by different research questions, despite conceptual similarities.
>
>
> We were also contacted by Liu et al. authors and we acknowledged their prior work, shared with them the updated manuscript, and committed to explain this situation during the ICLR rebuttal phase.
>
>
> We acknowledge that a more thorough related work search could have identified this earlier. However, we respectfully request that this is recognised as a case of independent work and not a breach of research integrity, as we have sought to address it promptly through revision as soon as we became aware of the overlap.
>
> **Clarification on the overlap and differences**
>
> We acknowledge that our metric Cover@τ is mathematically similar to G-Pass@k (Liu et al., 2024), which we discovered only after submission. Given the prior work, we agree that the novelty of the metric’s functional form is limited. However, we argue that our motivation and use of the metric is different. We use Cover@τ to contrast it to Pass@K as a tool for assessing reasoning boundaries.
>
>
> Similarities:
> - both metrics are a thresholded generalisation of Pass@k
>
>
> Differences:
> - motivation: crossover in Pass@K curves between the base and RLVR models has been used to argue that RLVR sharpens the reasoning capabilities of the base models, so we became interested in whether this effect is due to Pass@k; the work of Liu et al. predates the sharpening vs discovery debate and so their focus was more on assessing model stability (reliability threshold>0.5)
> - formulation: our metric assumes that tasks have different per-trial success probabilities pi, estimated directly from the n completions of each task, eliminating the need to set a separate k parameter.
> - connection to both Pass@K and Pass@1: In addition to the Pass@K connection, we highlight the connection between Cover@τ and Pass@1 and show that the Pass@1 is the AUC of Cover@τ
> - curve as a diagnostic tool for breadth-depth: we make the case for visually inspecting Cover@τ curve to assess whether a model has coverage (breadth) or whether it’s more stable (depth)
> - trade-offs revealed by Cover@τ that Pass@k may hide: we show that rankings based on Cover@τ imply the same rankings based on Pass@k, but not the other way around. This shows that Cover@τ curves reveal ordering differences across reliability levels that the Pass@k curves can hide.

---

> > ### Comment · Reviewer_Nwpw · 2025-11-27
> >
> > Thank you for the clarification. However, the rebuttal does not resolve my main concern. The claimed difference in motivation is not substantial. Both Cover@τ and G-Pass@kτ were proposed to address the same underlying limitation of Pass@k and to provide a more fine-grained view of RLVR reasoning ability. The mathematical structure, the use of a reliability threshold τ, and the interpretation are highly similar.
> >
> > The additional distinctions you mentioned: direct estimation of per-task success probabilities, notation choices, or visualization arguments, are minor variations rather than methodological innovations. As a result, the core contribution of the paper remains largely overlapping with Liu et al. (ACL 2025), and the novelty is limited.
> >
> > While this may be a case of independent rediscovery rather than intentional misconduct, the lack of citation and the strong conceptual overlap indicate insufficient literature review and improper attribution in the original submission.
> >
> > I therefore maintain my Strong Reject recommendation based on lack of originality, regardless of intent.

---

> > > ### Author Response · Authors · 2025-11-27
> > >
> > > Thank you for your feedback and for remaining engaged with our work throughout this review process.
> > > We updated the submission such that our claims and contextualisation within the prior work fully acknowledges the work of Liu et al. (2024), which we regretfully missed. We reiterate we proceeded similarly with the preprint version of this paper immediately on finding about the prior work, even before receiving these reviews.  We hope our actions further address your concerns regarding improper attribution.
> > >
> > > Next, while we agree, in hindsight, that the existence of G-Pass@k diminishes the novelty of the metric we devised, the empirical contribution of the paper can still be valuable to researchers and practitioners. We evaluate recent GRPO/PPO algorithms within a rigorous experimental design, involving 30 training runs on 6 sets, with out-of-distribution evaluation splits in order to illustrate the properties of Cover@τ and assess the models' reasoning performance and compare them on the breadth-depth spectrum, which to our knowledge, has not been been established.
> > >
> > > While we understand the reviewer may weigh the conceptual overlap strongly, we consider the empirical part of the paper has merits on its own and should not be overlooked when assessing usefulness or even novelty.

---

### Official Review · Reviewer_QhEP · 2025-11-01

**Soundness:** 2
**Presentation:** 3
**Contribution:** 2
**Rating:** 4
**Confidence:** 4

**Summary:**

This paper proposes Cover@τ, a new evaluation metric for assessing reasoning abilities of LLMs that accounts for reliability thresholds. The authors argue that Pass@k at large k is misleading for tasks with discrete answer spaces, as it conflates lucky guesses with genuine reasoning. They demonstrate that Pass@k is a Beta(1,k)-weighted average of Cover@τ, introduce the OMEGA and Reasoning Gym benchmarks, and evaluate several RLVR methods showing different breadth-depth trade-offs.

**Strengths:**

1. **Clear problem formulation**: Effectively articulates the degeneracy of Pass@k
   at large k with concrete examples (Figure 1), showing how models can achieve
   Pass@k=1 through random guessing on tasks with small answer spaces. The Beta(1,k)-weighted interpretation of Pass@k
   (Proposition 1) provides valuable insight into why Pass@k is biased toward low-τ
   regions, emphasizing "lucky hits" over reliability. Proposition 2 shows that
   Cover@τ dominance implies Pass@k dominance but not vice versa.

2. **Breadth-depth framework**: The explicit trade-off between coverage (low τ) and
   reliability (high τ) is conceptually clean and practically interpretable.

3. **Systematic RLVR evaluation**: First comprehensive comparison of RLVR methods
   (GRPO, PPO, GSPO, KL-Cov, Unlikeliness) using reliability-aware metrics, revealing
   that entropy-preserving methods (KL-Cov) achieve better stability.

**Weaknesses:**

1. **Insufficient acknowledgment of prior work**: The recent work "Are Your LLMs
Capable of Stable Reasoning?" (Liu et al., arXiv:2412.13147, Dec 2024) proposed
G-Pass@k, which measures the same concept—coverage at reliability threshold τ.
While your Cover@τ provides cleaner theoretical formulation and focuses on RLVR
methods, the submission should:
- Discuss the relationship between Cover@τ and G-Pass@k
- Clarify whether this is concurrent/independent work
- Compare the estimation approaches (your implicit empirical frequency vs. their
  explicit hypergeometric distribution)
).
2. **Shallow RLVR analysis**:
Why does preventing entropy collapse (KL-Cov) improve Cover@τ specifically, how do training hyperparameters affect the breadth-depth trade-off, what Cover@τ profile should practitioners target during training?
3. **Missing critical experiments**: Data contamination/overfitting effects on Cover@τ (present in G-Pass@k)
Sample complexity analysis: how does estimation variance change with n and k?
Ablation on AUC+ vs. alternative aggregation methods
4. **Limited practical guidance**: The paper shows Cover@τ reveals different rankings but doesn't provide actionable recommendations for τ selection based on application requirements.

**Questions:**

See the questions in **Weaknesses** part

---

> ### Author Response · Authors · 2025-11-24
> **Response (part 1/2)**
>
> Thanks for raising many important issues.
>
> **Relationship to prior work:**
>
> We started working on this paper in early 2025, where the Cover@τ metric arose in the context of the sharpening vs. discovery debate on RLVR’s role, introduced by the paper ‘Does Reinforcement Learning Really Incentivize Reasoning Capacity in LLMs Beyond the Base Model?’ by Yue et. al. (2025). At the time, we were not yet aware of the G-Pass@k metric of Liu et al. (2024), which was introduced earlier and focuses more on model stability. We were thus motivated by different research questions, despite conceptual similarities.
>
> Our ICLR submission was also uploaded as a preprint manuscript. Once we discovered the prior work, we uploaded an updated preprint version, in order to i) properly cite Liu et al. ii) clarify the overlap but also the difference in formulation and intended use.
>
> **Overlap and differences between Cover@τ and G-Pass@k**
>
> We acknowledge that our metric Cover@τ is mathematically similar to G-Pass@k (Liu et al., 2024), which we discovered only after submission. Given the prior work, we agree that the novelty of the metric’s functional form is limited. However, we argue that our motivation and use of the metric is different. We use Cover@τ to contrast it to Pass@K as a tool for assessing reasoning boundaries.
> Similarities:
>  - both metrics are a thresholded generalisation of Pass@k
>
> Differences:
>  - motivation: crossover in Pass@K curves between the base and RLVR models has been used to argue that RLVR sharpens the reasoning capabilities of the base models, so we became interested in whether this effect is due to Pass@k; the work of Liu et al. predates the sharpening vs discovery debate and so their focus was more on assessing model stability (reliability threshold>0.5)
>
> - formulation: our metric assumes that tasks have different per-trial success probabilities pi, estimated directly from the n completions of each task, eliminating the need to set a separate k parameter.
>
> - connection to both Pass@K and Pass@1: In addition to the Pass@K connection, we highlight the connection between Cover@τ and Pass@1 and show that the Pass@1 is the AUC of Cover@τ
> - curve as a diagnostic tool for breadth-depth: we make the case for visually inspecting Cover@τ curve to assess whether a model has coverage (breadth) or whether it’s more stable (depth)
> - trade-offs revealed by Cover@τ that Pass@k may hide: we show that rankings based on Cover@τ imply the same rankings based on Pass@k, but not the other way around. This shows that Cover@τ curves reveal ordering differences across reliability levels that the Pass@k curves can hide.
>
> We will also revise the Openreview manuscript to properly cite the work of Liu et al (2024) and include the discussion about overlap and differences to GPass@k.
>
> **Why do entropy mitigation methods help cover@tau specifically? How do training hyperparameters affect the breadth-depth trade-off?**
>
> Our goal is to introduce cover@tau as a diagnostic and uncover breadth-depth properties that pass@k may obscure. We provide comparisons on some of the representative methods in RLVR and illustrate pass@1 versus cover@tau for relevant tau values for breadth and depth. We agree that a more thorough investigation into how different algorithms and specific training schemes affect breadth and depth is a valuable future direction.
>
> To better understand why KL-cov performs well under cover@tau, we plot the per-task success probabilities pi. We show a scatter plot where the pi values of GRPO are on the Ox axis and the pi values of KL-cov on the Oy axis (link to plot: https://files.catbox.moe/ekiymt.png ). The plots illustrates two things:
>  - breadth:  for many tasks where GRPO has near-zero success probability (pi~=0), KL-cov uplifts them softly (which increases breadth)
>
>  - depth: for many tasks where GRPO has good success probabilities pi>0.5, KL-cov increases them further, resulting in increased reliability at high tau values
>
> Together, these shifts explain why KL-cov dominates GRPO across the entire reliability spectrum. Pass@1 is the AUC of cover@tau, so it reflects the overall improvements in the per-task success probabilities, but does not show how these improvements are distributed across different reliability levels. However, cover@tau showcases whether the gains come from increased performance at low tau levels (breadth), high tau levels (depth), or both.

---

> ### Author Response · Authors · 2025-11-24
> **Response (part 2/2)**
>
> **Sensitivity (how does estimation variance change with n and k?)**
>
> In our formulation, each task has a success probability pi, and we define cover@tau as the fraction of tasks with pi>=tau. In practice, we sample n completions and we estimate pi directly as the number of correct outputs out of the n completions (pi=c/n). Thus, the only relevant parameter affecting estimation variance in our setting is n, the number of sampled completions; there is no separate subset-size parameter k in Cover@τ.
>
> (Clarification: in the main paper, we used the symbol “k = 8192” to denote the number of completions per task. In this rebuttal, we use the standard notation n for this quantity to avoid confusion with the subset-size k used in Pass@k and G-Pass@k.)
>
> To evaluate sensitivity to n, we picked a GRPO checkpoint and recomputed Cover@τ curves on a subset from the Omega dataset using n ∈ {256, 512, 1024, 2048, 4096} samples per task (link to plot here: https://files.catbox.moe/gutw5d.png). For smaller n values (256, 512), cover@tau values show slightly increased noise, but for n ≥ 1024, the curves are almost indistinguishable, showing that the metric is robust to n.
>
> **Concerns about data contamination/overfitting effects on Cover@tau**
>
> We agree that the contamination/overfitting effects analyzed by Liu et al. (2024) are valuable, since memorizing can artificially uplift the per-task success probabilities. However, our primary goal was to introduce Cover@τ as a reliability-aware tool and evaluate breadth-depth capabilities in a setup which disentangles contamination from reasoning performance. Therefore, we chose Omega and Reasoning Gym datasets, which were designed to probe OOD generalization and reduce memorization issues, so reproducing contamination experiments would be less informative in our setting.
>
> **Practical guidance**
>
> We agree that actionable recommendations are important for practitioners.
>
> We view the full cover@tau curve as an exploration tool, useful for inspecting the entire reliability profile and identifying exactly where curves cross between two models.
>
> Point evaluations provide comparisons at specific reliability levels. We provided tau = 0.2 and tau = 0.8 to assess breadth-depth as they illustrate two ends of the reliability spectrum, but they are not definitive values and other choices can be made based on application needs.
>
> Choosing tau values based on reliability:
>  - cover@tau for low tau values (e.g. 0.1-0.3) reflects breadth capabilities (‘how many tasks does my model solve at least sometimes?’); these values are also useful for picking models where creativity or partial success is valuable (searching for research ideas)
>  - cover@tau for high values (e.g. 0.7-0.9) reflects depth (‘how many tasks does my model consistently solve?’); these values are also useful for picking models where there is less room for errors (code generation)
>
> We refrained from prescribing specific values, as practitioners may need to select tau based on the different constraints of their problems. If a single value is required, a choice for tau should be made depending on the risk and reliability imposed by the problem (lower tau values for problems where errors are tolerated and higher tau values for problems where consistency is key).

---

### Official Review · Reviewer_FVku · 2025-11-01

**Soundness:** 3
**Presentation:** 3
**Contribution:** 3
**Rating:** 4
**Confidence:** 4

**Summary:**

The paper argues that Pass@k at large k can be misleading for reasoning tasks with discrete answer spaces, because success can arise from random guessing without reliable reasoning. The authors propose Cover@τ, the fraction of problems for which at least a τ proportion of sampled completions are correct. They prove that Pass@k is a Beta weighted average of the Cover@τ curve, so Pass@k emphasizes very low τ and ignores reliability, and they also introduce a pairwise comparison measure AUC⁺_cover. Experiments on OMEGA and Reasoning Gym show that rankings of RLVR methods change when we look at coverage under reliability thresholds, with methods that preserve exploration and entropy doing better at higher τ. The crossover plots and the tables in this paper support the claim that Cover@τ reveals different capability trade-offs than Pass@k.

**Strengths:**

1. Cover@τ provides a simple but powerful reliability controlled view. The identity that Pass@k is a Beta weighted integral over τ makes the bias of large k very clear and easy to communicate to practitioners. This theory part is clean and correct from first principles.
2. On OMEGA Probability, the figure on page 2 shows Pass@k saturates as k grows because the answer support is small, while the Cover@τ curve decreases smoothly and separates models by reliability.
3. The work addresses the widely reported crossover between RLVR and base models under Pass@k and provides a principled reason why this happens.

**Weaknesses:**

1. To estimate per-problem success rates, the paper uses very large K, up to 8196 samples, but the statistical uncertainty of the empirical proportions is not analyzed. Confidence bands for Cover@τ and sensitivity to K and temperature are important for fair comparisons, especially when τ is small.
2. All main results are on math with numeric answers. Code tasks with test suites and other verifiable domains would strengthen the claim that the metric generalizes across RLVR use cases. The ongoing literature already evaluates both math and code when discussing reasoning boundaries.
3. Experiments start from a single family and size, Qwen 2.5 7B Instruct, so it is hard to know if the findings hold for other families or larger models. For example, due to contaminations, Qwen2.5 7B may perform very differently from Llama. Several reports show that behavior under RLVR depends on the base model and training recipe.

**Questions:**

1. How sensitive are your Cover@τ curves to the sample size K, decoding temperature, and nucleus top p. Can you add bootstrap confidence intervals or a Bayesian treatment that reports uncertainty on coverage across τ.
2. Can you include a code generation study, for example HumanEval or other verifiable suites, and compare Cover@τ with CoT Pass@k and maj@k there. This would help show whether the metric behaves similarly beyond numeric answers.
3. Could you report results on at least one more base model family or size. Some recent papers suggest that the interplay between RLVR and base distribution varies across models.

My final rating depends on the rebuttal.

---

> ### Author Response · Authors · 2025-11-24
> **Response**
>
> Thank you for your feedback. We agree that quantifying uncertainty and sensitivity is important.
>
> **Sensitivity to the number of generated samples (n):**
>
> *Clarification: in the main paper, we used the symbol 'k' to denote the number of completions per task. In this rebuttal, we use the standard notation 'n' for this quantity to avoid confusion with the subset-size k used in Pass@k and G-Pass@k, which our metric doesn’t use.
> We picked a PPO checkpoint and recomputed Cover@τ curves on the algebra_func_intersection OOD subset from the Omega dataset using n ∈ {256, 512, 1024, 2048, 4096} samples per task (link to image here https://files.catbox.moe/gutw5d.png). For smaller n values (256, 512), cover@tau values show slightly increased noise, but for n ≥ 1024, the curves are almost indistinguishable, showing that the metric is robust to n.
>
> **Sensitivity to temperature**
>
> We keep n fixed (4096) and sample with different temperatures during inference (t=0.6, t=0.8 and t=1.0) from models trained on arithmetic_gcd train set from Omega and evaluate them on the arithmetic_gcd OOD test set. We evaluate GRPO, PPO (GAE) and KL-cov methods, trained from the Qwen2.5-7B-Instruct base model. Image plots are listed here ([link temp 0.6](https://files.catbox.moe/2hqowg.png), [link temp 0.8](https://files.catbox.moe/3nk1o6.png), [link temp 1.0](https://files.catbox.moe/mxbglp.png). We notice that the cover@tau curves are relatively stable for the 3 models across the temperatures.
>
> **Bootstrapped confidence intervals**
>
> We compute bootstrapped CI for cover@tau over 280 Omega OOD tasks (spanning the 5 subsets GCD, Function Intersection, Probability No Fixed, Digit Sum, and Circle), by sampling with replacement the per-task success probabilities. The link to the plot is here https://files.catbox.moe/0qfukr.png. GRPO and PPO (GAE) curves have overlapping confidence bands, indicating that at this dataset size, cover@tau can’t distinguish fine-grained performance rankings between these two methods. However, the KL-cov cover@tau curve sits above all the other curves, with non-overlapping bands, indicating robust advantage in both breadth (lower tau) as well as depth (higher tau), confirming the results in our paper. The base model’s curve indicates the weakest performance across the entire reliability spectrum.
>
> **Sensitivity to model family**
>
> We trained RLVR models starting from another base model, LLama3.1-8B. We trained them on 2 subsets from Omega and one set from Reasoning Gym. We will update with the results.
>
> **pass@k behavior wrt non-numerical answers and other domains**
> While we did illustrate the ‘lucky hits’ argument for pass@k for test sets with numerical answers and small support (Omega), the other dataset that we tried (Reasoning Gym) has answers in the form of mathematical symbolic expressions (integrals), which provide a larger answer space. We include here an additional plot where pass@K and cover@tau curves are illustrated on the Reasoning Gym test set (link here https://files.catbox.moe/sqjylz.png) The crossover here is still present, but less extreme: at large K (4096), the base model is almost on par with the PPO model in terms of pass@K. The cover@tau curves illustrate that the base model has less coverage than almost all the other RLVR methods at both low-tau (breadth) and high-tau (depth) regions. We agree that experiments for code generation would strengthen the conclusions. The crossover phenomenon between the base and RLVR models has also been observed in code (Figure 4 in Yue et al. https://arxiv.org/pdf/2504.13837). These results suggest that the ‘lucky hits at large k’ behavior is not limited to numeric answers with small support, it also appears for symbolic answer spaces, where the support is larger but still structured.

---

> > ### Author Response · Authors · 2025-11-28
> >
> > **Experiments on a different model family**
> >
> > We ran preliminary experiments by choosing Llama-3.1-8B-Instruct as the base model. We trained GRPO, KL-cov and GSPO models on 2 OOD sets of the Omega dataset (arithmetic_gcd and arithmetic_mixed). Links to the 2 plots here: [arithmetic_gcd](https://files.catbox.moe/407khr.png), [arithmetic_mixed](https://files.catbox.moe/jdnc5s.png).
> >
> > Early results confirm observations consistent with our Qwen results: the base model exhibits similarly large pass@K performance at large K, overtaking some of the RLVR models, while its cover@τ  performance is the lowest on the entire reliability spectrum.
> >
> > The cover@τ  curves also indicate better performance of GSPO wrt the other models at both lower τ values (0.2) and higher τ  values (0.8), but more train runs on other sets are required to establish fine-grained performance rankings between the RLVR models.

---

### Author Response · Authors · 2025-11-27
**updated manuscript**

We updated the manuscript to include the missing citations, including Yue et al. (2025) and Liu et al. (2024), and added a discussion in the Related Work section clarifying the relationship and overlap between Cover@τ and G-Pass@k. All additions are highlighted in blue for ease of review.

---

### Meta-Review · Area_Chair_hUNn · 2026-01-07

**Summary:**

This paper introduces Cover@τ, a metric measuring the fraction of problems for which at least a τ proportion of completions are correct, intended to address limitations of Pass@k when evaluating reasoning boundaries of RLVR models. The reviewers acknowledge the paper addresses a relevant problem with clear theoretical exposition. However, a critical issue undermines the contribution: the proposed metric is mathematically equivalent to G-Pass@k from Liu et al. (2024), which was not cited in the original submission. Although the authors claim independent development and have since updated the manuscript to acknowledge this prior work, the core methodological novelty is substantially diminished. The empirical contributions, while systematic, are insufficient to compensate for this overlap. Additional concerns regarding limited experimental scope (primarily one model family, math-only tasks) and insufficient practical guidance for τ selection were only partially addressed.

**Reviewer Concerns:**

The authors adequately addressed several technical concerns through additional experiments: sensitivity analyses for sample size and temperature, bootstrapped confidence intervals, preliminary experiments on Llama-3.1-8B, and random guess baseline experiments demonstrating problematic Pass@k behavior on numeric answer tasks.

However, the fundamental concern regarding overlap with G-Pass@k remains outstanding. Reviewer Nwpw argued that the claimed motivational and formulation differences are minor variations rather than methodological innovations, and explicitly maintained their Strong Reject after the rebuttal. While the authors updated the manuscript to cite Liu et al. and discuss differences, the distinctions (direct estimation of success probabilities, visualization approach, breadth-depth framing) do not constitute sufficient novelty for a top venue. Code generation experiments requested by multiple reviewers were not provided, and main results still rely primarily on a single model family.

**Reviewer Scores:**

Reviewer FVku (Score: 4): Sensitivity and uncertainty analyses were provided; code generation experiments remain missing. Score would likely remain unchanged.

Reviewer QhEP (Score: 4): The prior work acknowledgment concern was addressed through manuscript revision, but the reviewer's view on limited novelty would likely persist. Score would likely remain unchanged.

Reviewer Nwpw (Score: 0): Explicitly maintained Strong Reject after the rebuttal, stating the novelty concern remains unresolved. Score unchanged.

Reviewer uWMw (Score: 4): Expressed willingness to increase score after authors provided random guess experiments, but maintained concerns about limited novelty. Score would likely increase modestly.

---

### Decision · Program_Chairs · 2026-01-26

Reject